# Identification of the RPGR Gene Pathogenic Variants in a Cohort of Polish Male Patients with Retinitis Pigmentosa Phenotype

**DOI:** 10.3390/genes14101950

**Published:** 2023-10-17

**Authors:** Katarzyna Nowomiejska, Katarzyna Baltaziak, Paulina Całka, Marzanna Ciesielka, Grzegorz Teresiński, Robert Rejdak

**Affiliations:** 1Chair and Department of General and Pediatric Ophthalmology, Medical University of Lublin, 20-059 Lublin, Poland; k.baltaziak@gmail.com (K.B.); robertrejdak@yahoo.com (R.R.); 2Department of Forensic Medicine, Medical University of Lublin, 20-059 Lublin, Poland; paulina.calka@umlub.pl (P.C.); marzanna.ciesielka@umlub.pl (M.C.); grzegorz.teresinski@umlub.pl (G.T.)

**Keywords:** retinitis pigmentosa, RPGR gene, next-generation sequencing

## Abstract

The goal of the study was to explore the spectrum of pathogenic variants in the RPGR gene in a group of male Polish patients with a retinitis pigmentosa (RP) phenotype. A total of 45 male index patients, including twins, being members of 44 families, were screened for pathogenic variants in the RPGR gene via the direct sequencing of PCR-amplified genomic DNA and underwent a comprehensive ophthalmological examination in one center located in Poland. A total of two pathogenic and five likely pathogenic variants in eight patients (18%) were detected in the studied cohort. Of these, five variants were novel, and five disease-causing variants (71%) were identified within the ORF15 mutational hotspot of the RPGR gene. The median age of onset of the disease was 10 years (range 6–14 years), the median age during the examination was 30 years (range 20–47 years), and the median visual acuity was 0.4 (range 0.01–0.7). The majority of patients had middle constriction of the visual field and thinning of the central foveal thickness. Dizygotic twins bearing the same hemizygous mutation showed a different retinal phenotype in regard to the severity of the symptoms. This is the first RPGR mutation screening in Poland showing a prevalence of 18% of RPGR pathogenic mutations and likely pathogenic variants in the studied cohort of male patients with an RP phenotype.

## 1. Introduction

Retinitis pigmentosa (RP) is a highly heterogeneous group of retinal dystrophies and is clinically characterized by initial nyctalopia and progressive visual field constriction that may culminate in blindness [1]. RP can be inherited in an autosomal dominant (adRP), autosomal recessive (arRP), or X-linked (XLRP) manner, with rare digenic and mitochondrial forms [2]. To date, mutations in 23 genes are known to cause adRP, mutations in 44 genes cause arRP, and mutations in 3 genes cause XLRP [3]. XLRP can be caused by disease-causing variants in the *RPGR*, *RP2*, and *OFD1* genes, with RPGR being the most prevalent among XLRP [4].

Disease-causing variants in the RPGR (retinitis pigmentosa GTPase regulator) gene account for 70–90% of X-linked retinitis pigmentosa (XLRP) [5,6] and 10–20% of all RP familial cases [6], which is higher than most other single RP loci [7]. Thus, it seems that RPGR disease-causing variants are one of the most common causes of retinal degeneration in addition to rhodopsin (RHO) gene disease-causing variants [8]. XLRP is associated with a severe phenotype in terms of onset and progression of the disease, starting as early as the first decade of life and progressing into complete blindness by the third or fourth decade [9,10]. Due to X-linked inheritance, the disease predominantly affects males. The RPGR gene (OMIM 312610) is located in the Xp21.1 chromosomal region and is composed of 23 exons, including the alternatively spliced exons 9a, ORF15, 15a, and 15b [5,11]. RPGR localizes to the connecting cilium of photoreceptors and is believed to play a role in protein transport [12,13]. Approximately 60% of disease-causing variants in RPGR are found in an additional, alternatively spliced C-terminal exon, called open reading frame (ORF) 15, which is highly expressed in photoreceptors [5]. ORF15’s sequence is highly repetitive and purine-rich and is a hotspot for disease-causing variants [5]. Exon ORF15 is predominantly expressed in the retina and brain, whereas mRNAs, including exons 1–19, are widely expressed [14]. In humans, out-of-frame deletions, duplications, or insertions are frequently found in ORF15, whereas nonsense disease-causing variants are rare, and disease-causing missense variants have not been described. Next-generation sequencing (NGS) has increased the number of variants identified in RPGR in recent years.

The Polish population of RP patients has not been screened exclusively for RPGR gene disease-causing variants so far. Reports of pathogenic variants in the Polish population of patients with inherited retinal dystrophies (IRDs) in the literature are scarce [15,16,17,18].

The aim of this study was to identify the pathogenic variants of the RPGR gene in a cohort of Polish male patients with a phenotype of RP.

## 2. Material and Methods

Approval from the ethics committee of the Medical University of Lublin has been obtained (nr KE-0254/343/2018). The study was performed in accordance with the tenets of the Declaration of Helsinki, and informed consent was obtained from each individual tested or from parents or guardians of individuals under age 18.

### 2.1. Genetic Testing

All probands were naïve in regard to genetic testing. Blood samples (5 mL EDTA) were collected in order to perform molecular genetic testing in the Department of Forensic Medicine of the Medical University of Lublin, Poland. Patients’ DNA was isolated from the peripheral blood using Blood Miniprep kits (A&A Biotechnology, Gdansk, Poland), and an AmpliSeq-based kit designed in DesignStudio v3.0 software (Illumina, San Diego, CA, USA) was used for RPGR gene analysis.

The entire RPGR gene, from location chrX 38128416 to 38186817, was sequenced. The in silico coverage of the entire gene was 94.5%. The location chrX 38145032–38145785 was not covered by NGS sequencing, i.e., 753 bp of the ORF15 region; therefore, the whole ORF15 was sequenced using the Sanger method. We used primers described in the publication of Neidhardt and colleagues [11]. In the first stage, long-range PCR, which covers the entire ORF15 region, was performed using TaKaRa LA Taq (Clontech, Mountain View, CA, USA). Approximately 200 ng of DNA was used in a final volume of 50 µL. The PCR conditions for RPGR-ORF15 were initial denaturation at 94 °C for 2 min, followed by 45 cycles: denaturation at 98 °C for 10 s, primer annealing at 60 °C for 1 min, extension at 72 °C for 3.5 min and final extension at 72 °C for 10 min. The PCR product was treated with ExoSAP-IT (Affymetrix, Santa Clara, CA, USA) and sequenced using BigDye Terminator v3.1 Cycle Sequencing Kit (Applied Biosystems, Waltham, MA, USA) per the manufacturer’s protocol. The analysis was carried out using the SeqScape v.2.7 software based on the transcript: NM_001034853.2. Depth coverage in the ORF15 region ranged from min 151× to max 930×. Variants identified by the NGS technology in the ORF 15 region were confirmed via direct sequencing.

Amplification of DNA fragments was carried out in two pools: in one, there were 108, and in the other, 109 amplicons with an average length of 350 pz. NGS sequencing was conducted on a MiSeq platform (Illumina, San Diego, CA, USA) using MiSeq Reagent Kit v3 reagents. An average coverage of 150× was obtained for the designed panel. The obtained data were compared to the reference UCSC hg38 genome. MiSeq Reporter v2.6 software was used to analyze the data, followed by the Galaxy platform and FASTQC, BWA-MEM, and Free Bayes algorithms. The potential pathogenicity of the obtained variants was verified using the resources of public databases such as dbSNP, ClinVar, Varsome, gnomAD, RefSeq, Ensembl, Franklin genoox, and VarSeak.

### 2.2. Clinical Data Collection

The patients were recruited at the Chair and Department of General and Pediatric Ophthalmology of the Medical University of Lublin, Poland. A total of 45 male patients have been enrolled in the study. Inclusion criteria were as follows: clinical symptoms of RP, male gender, and Polish origin. The majority of patients (25) came from families with two or more affected male relatives, 9 patients had a family history with at least two generations of affected males that were related through an unaffected or carrier female, 11 patients were the only affected males in the family with RP but were given the tentative diagnosis of XLRP on the basis of clinical presentation (e.g., early onset and severity of disease). This subdivision of patients reflected the strength of the clinical documentation and the family history of the disease, similar to a study by Breurer [6]. Patients with a family history of parental consanguinity were not included. The diagnosis of RP was made in each individual on the basis of an ophthalmological examination.

A detailed medical and family history was taken from each patient to create pedigrees. The ophthalmological evaluation included best-corrected Snellen visual acuity (BCVA), visual field testing with Octopus 900 perimeter (Haag Streit, Koeniz, Switzerland), dilated ophthalmoscopy, digital wide-field fundus photography (Optos, Dunfermline, UK), wide-field autofluorescence imaging (Optos, Dunfermline, UK), spectral-domain optical coherence tomography (SD-OCT, Topcon, Tokyo, Japan), and electrophysiological assessment (Tomey, Nagoya, Japan). Central foveal thickness obtained in OCT was the mean thickness at the point of intersection of 6 radial scans.

## 3. Results

### 3.1. Genetic Testing Results

Among all 45 male patients, we identified seven distinct variants in eight patients (including male twins) in the RPGR gene, including five variants present in ORF15. Among all the variants, five were novel: two duplications and three single-nucleotide variants (SNV), two deletions have already been described in the literature (Table 1). These variants resulted in a frameshift in four variants, two missense mutations, and one nonsense mutation.

The variants described in ClinVar as pathogenic were not analyzed further. The other variants were classified using the bioinformatics software: Mutation Taster, Varsome, Franklin genoox, and SIFT. The missense mutations were additionally analyzed by Varseak and polyPhen2 tools. In cases of missense mutations, in silico functional analysis was performed using Polyphen2 software (Polymorphism Phenotyping v2). In silico analysis (using varSEAK, PolyPhen-2, and SIFT) was performed for two patients with a missense mutation. These changes do not affect splicing but can impact protein structure and function. Obtained values (PolyPhen-2—score: 1“probably damaging”; SIFT—score: 0.01 deleterious) corroborate their potential pathogenicity.

Electropherograms of NGS variants identified in ORF15 confirmed by Sanger sequencing are presented in Figure 1. Variants c.593G>T (patient VII) and c.799G>C (patient VIII) are outside the ORF15 region identified in NGS technology: variant depth 486× (99%) and 389× (99%), and they have not been confirmed in the Sanger analysis.

### 3.2. Pedigrees

Among seven families with confirmed pathogenic or likely pathogenic variants, there was one family (family 4) with male twins and the same variant (Figure 2).

### 3.3. Clinical Examination Results

The main clinical findings of the cohort are included in Table 2. The median age of the eight patients with disease-causing variants in the RPGR gene at the onset of the disease was 10 years (range 6–14 years). The median BCVA was 0.4 (range 0.01–0.7), and the median age at examination was 30 years (range 20–47 years). Mean central foveal thickness was 221 µm (range 143–260 µm).

The majority of the patients (six patients apart from the twins) presented with a typical RP retinal phenotype with bone-spicule pigment deposits in the mid-peripheral retina, optic disc pallor, and RPE atrophy (Figure 3). Of these, six patients demonstrated middle constriction of the visual field, and two patients demonstrated advanced constriction of the visual field (10 degrees with V4e isopter).

Among the eight patients with disease-associated variants in the RPGR gene, there were two male dizygotic twins (patient IV and V, family 4) with the same variant c.2389dup. The phenotype of these twins differed with regard to the visual acuity, visual field, and FAF patterns (Figure 4 and Figure 5).

## 4. Discussion

This is the first study focused on reporting the pathogenic variants exclusively in the RPGR gene in a cohort of Polish male patients with a phenotype of RP. As a result, two pathogenic and five likely pathogenic variants in 8 index patients, including dizygotic twins bearing the same variant, were identified.

Currently, more than 500 pathogenic variants have been identified throughout the RPGR gene [20], and over 55% of these variants occur throughout the length of the photoreceptor-specific ORF15. It is already known that the relative prevalence of pathogenic variants varies somewhat between populations. Moreover, studies around the world on RPGR retinal diseases have shown marked genotypic and phenotypic heterogeneity. In a large study by Cehajic-Kapetanovic and colleagues [21] where they analyzed the data of a single cohort of 116 male patients from the genetic databases at two clinical centers (Oxford, United Kingdom, and Bonn, Germany), a total of 60 different pathogenic variants were identified in the ORF15 region of the RPGR gene. They found the commonest form of RPGR-related disease to be the rod–cone phenotype, also known as the “classic” X-linked RP phenotype, which was present in 56% of 60 patients (60/108) compared to 94% [22], 85% [23], or 70% [24] of the total number of patients observed in studies from Germany, China, and a European multicenter study, respectively. In our study, the percentage of detected variants in the ORF15 region was 71%, similar to other studies, for example, 73% in a study by Bellingrath and coworkers [21] and 66% in a study by Sharon and colleagues [25].

Tracewska and colleagues [15] have already reported on the prevalence of inherited retinal diseases in 190 Polish families. The conclusion was that the most prevalent cause of IRD in Poland was ABCA4-associated diseases, regardless of the phenotype. In Polish patients with RP, the second most prevalent causal gene was RHO, and the third was RPGR, while there were not as many disease-causing variants in the EYS gene as in Western populations. The X-linked recessive pattern was confirmed by finding causal variants in eight families, of whom six had variants in RPGR, one had a variant in CACNA1F, and one had a mutation in CHM. One additional isolated male patient appeared to have a pseudoheterozygous mutation in RP2, an X-linked gene. This change was not present in the mother and most probably represents an early embryonic somatic mutation present in the peripheral blood and the buccal swab-derived DNA of the patient. Wawrocka and coworkers analyzed the molecular genetic basis of cone–rod dystrophy in 18 unrelated families of Polish origin [26]. They identified five novel variants, including one variant found in ORF 15: c.3142_3143dupAA, p.(Glu1049Argfs*41). Sanger sequencing of the RPGR-ORF15 was performed in three families presenting with an X-linked mode of inheritance and in three additional families with the autosomal recessive X-linked mode of inheritance. Moreover, Bukowy-Bieryllo and colleagues described two male Polish patients with syndromic primary ciliary dyskinesia [27]. Additionally, one case report of a variant NM_001034853: c.2899delG (p.E967Kfs*122) has been reported in a Polish patient [28].

The RPGR gene has been associated with several disease patterns, including rod–cone dystrophy (RCD, 70%), cone–rod dystrophy (CORD) (6–23%), and cone dystrophy (COD) (7%) [24]. Among the 277 described RPGR ORF15 inherited retinal disease variants (data from the HGMDPro database [21]), only 47 were associated with COD and CORD.

In our study, all eight patients had the phenotype of RP, although five variants were located in ORF15 and two variants in exons 6 and 8. It has been suggested that variants in exons 1–14 and the proximal part of the ORF15 exon usually result in RP, while variants in the distal end of the ORF15 exon cause COD/CORD. No disease-causing variants have been identified in exons 16–19 so far. The highly repetitive and purine-rich nature of the ORF15 exon of RPGR complicates the examination of the ORF15 sequence [5,6]. ORF15 sequencing relies on traditional Sanger sequencing because of the technical difficulties, and only a few laboratories are able to screen for it [25]. It has been shown in animal models of XLRP that the frameshift mutation in the RPGR exon ORF15 causes photoreceptor degeneration and inner retina remodeling [29]. Moreover, the frameshift mutation dramatically alters the deduced amino acid sequence, and the protein aggregates in the endoplasmic reticulum of the transfected cell [30]. Out-of-frame deletions, duplications, or insertions are frequently found in ORF15, whereas nonsense disease-causing variants are rare and disease-relevant missense disease-causing variants have not been described [11]. We report four frameshift and one nonsense mutations in our series of 5 variants in ORF15, including three new variants within the ORF15 region: two frameshift and one nonsense mutations.

Identification of RPGR gene disease-causing variants illustrates the importance of NGS panel testing in patients suffering from IRDs. It is especially important in Poland, as there is no full reimbursement from the national health service for panel genetic testing in IRDs. Previous studies vary in the detection rates of RPGR disease-causing variants within male patients with RP from 0–32% [6,25]. Branham and colleagues [8] identified pathogenic disease-causing variants of the RPGR gene in 29 of 214 patients (12 in ORF 15) with simplex RP or CORD screened in male subjects of the United States, which accounts for 15%, similar to our study. A study of 127 French families included 25 isolated males suspected of XLRP based on early onset, rapid progression, and subnormal visual acuity. Of these, 4% of the patients had disease-causing variants in RP2, 4% in RPGR exons 1 through 14, and 24% in RPGR ORF15 [31]. In the study by Churchill and colleagues, RP and RPGR disease-causing variants were found in 8.5% of families (22 of 258 families) with a provisional diagnosis of autosomal dominant RP [19]. Neidhart and coworkers [11] analyzed 141 RP families with possible X-chromosomal inheritance. In total, they identified 46 families with pathogenic sequence alterations in RPGR and RP2, of which 17 disease-causing variants have not been described previously.

Clinical evaluation and interviews with the patients from this study showed that the course of the disease, with respect to age-of onset and visual acuity, varied in affected males. The age of onset of patients from our study varied from 6 to 12 years; thus, it was quite an early onset of the disease. Visual acuity ranged from 0.01 to 0.7; the majority of patients had quite good central visual acuity (more than 0.2), and only one patient had low visual acuity (0.01). The majority of patients had middle concentric constriction of the visual field obtained with kinetic perimetry using three or fewer isopters. Central foveal thickness obtained in OCT in our series of patients was below the mean value of 240 µm in normal subjects. In the retrospective longitudinal study, including 34 index patients from France with a molecular diagnosis of RPGR-related CD/CRDs, 15 subjects were affected by CD, while the remaining patients had CRD [32]. More than 60% of these patients had reached a visual acuity of less than 0.1 in their best eye after their fifth decade of life. However, all 27 pathogenic variants were located in ORF15, and the phenotype was quite severe. All the patients from the present study had a phenotype of RP, and the majority of them had visual acuity better than 0.2; thus, the deterioration of the visual function will probably not be as pronounced as in patients with COD and CORD phenotypes. Interestingly, there were dizygotic twins among our cohort of patients, and they exhibited different phenotypes with different visual acuity, visual fields, and FAF patterns. The differences in the retinal phenotype may be due to incomplete penetrance; thus, genotype–phenotype correlations may be very difficult. Moreover, other genes may be involved in the RP genotype; thus, whole-genome sequencing (WGS) could be helpful in understanding the complete genetic background. Genes affected in RP are numerous and cover different steps of the visual cycle (phototransduction, retinal metabolism, tissue development and maintenance, cellular structure, and splicing) [33].

Our study extends the spectrum of pathogenic variants in the RPGR gene and describes the results of mutational screenings in male patients with an RP phenotype in Poland. The limitation of this study is a small cohort of included patients related to the rarity of the condition. Results obtained in this prospective study may have substantive implications for the calculation of recurrence risk, genetic counseling, and potential treatment options in Poland and illustrate the importance of genetic testing. The knowledge of the underlying gene defect will be valuable to determine which trials a patient might be eligible for in the future. It has already been emphasized that the RPGR gene should be the first gene to be screened in male patients with an RP phenotype. We hope that more investigations regarding the RPGR gene in Poland will be available in the near future.

## Figures and Tables

**Figure 1 genes-14-01950-f001:**
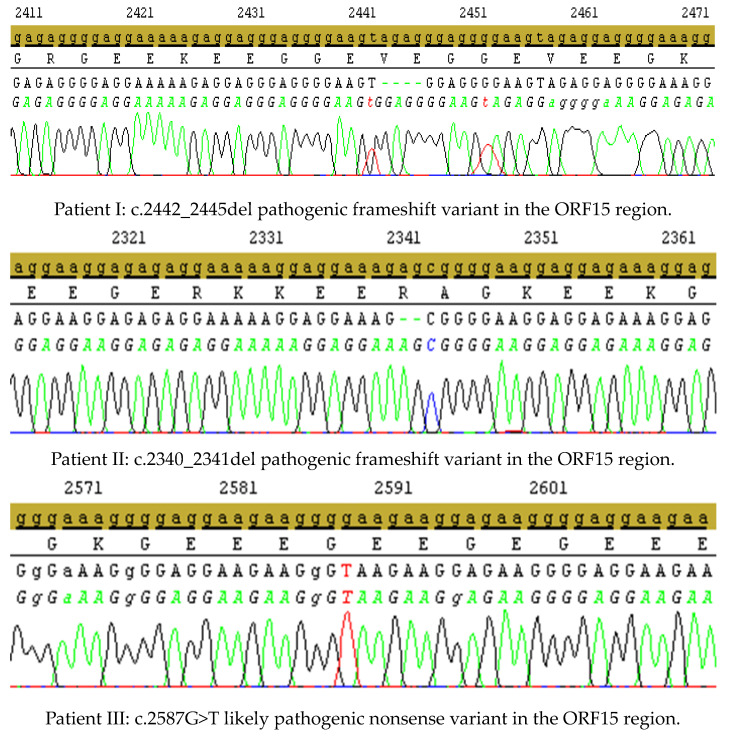
Electropherograms of the variants detected in this study in the ORF 15 region: patients I–VI (patients IV and V are twins).

**Figure 2 genes-14-01950-f002:**
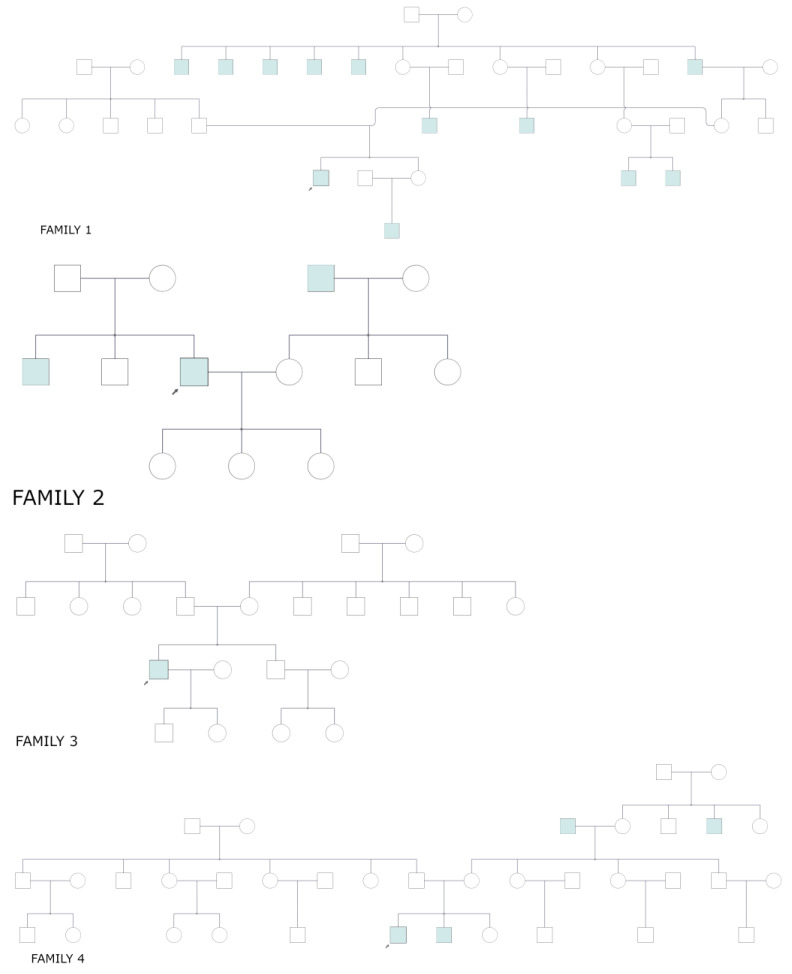
Pedigrees of the seven families of the eight index patients with pathogenic and likely pathogenic variants in the RPGR gene. Family 4 includes dizygotic twins. Arrow means index patient.

**Figure 3 genes-14-01950-f003:**
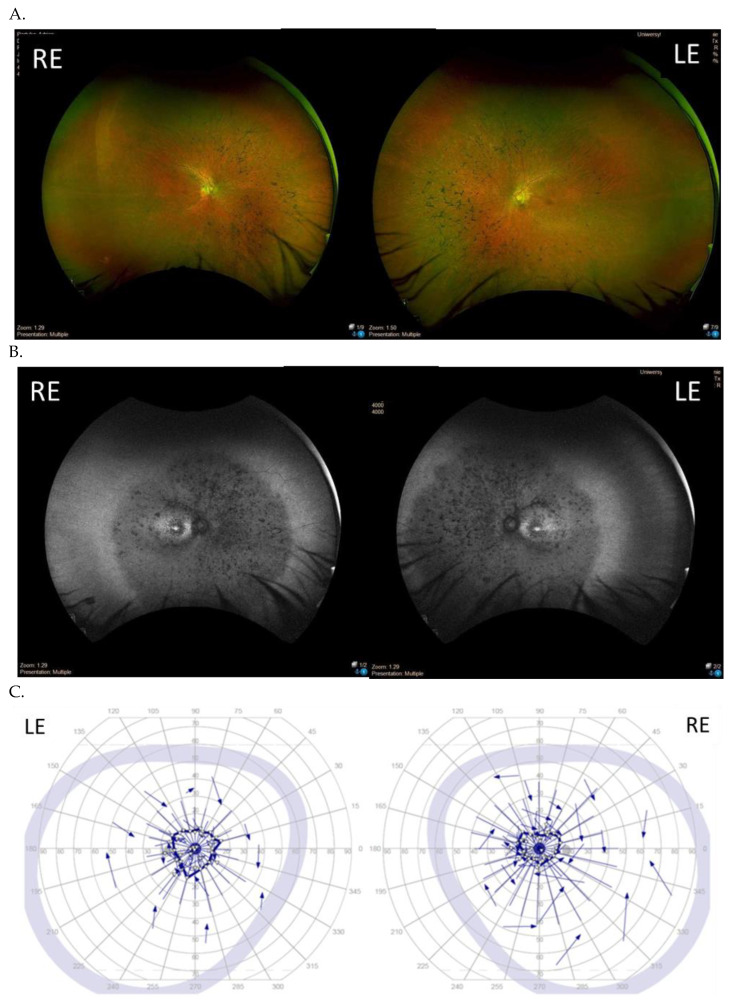
Results of the clinical ophthalmological examinations of both eyes of one patient with the pathogenic variant c.2442_2445del in ORF15 (family 1, patient I). Visual acuity is 0.7 in the right eye (RE) and 0.7 in the left eye (LE). From top to bottom: Optos wide-field color fundus photography (**A**), fundus autofluorescence (**B**), visual field obtained with V4e isopter (**C**), and optical coherence tomography of the macula (**D**).

**Figure 4 genes-14-01950-f004:**
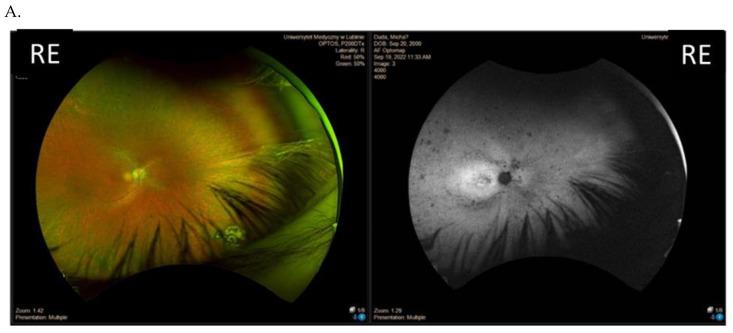
Results of the clinical ophthalmological examinations of both eyes of one patient with a c.2389dup likely pathogenic variant in ORF15 (family 4, patient IV, one of twins). Visual acuity is 0.7 in both eyes. From top to bottom: Optos wide-field color fundus photography and fundus autofluorescence of the right eye (RE), left eye unavailable (**A**), visual field obtained with V4e and III4e isopters (**B**), and optical coherence tomography of the macula (**C**).

**Figure 5 genes-14-01950-f005:**
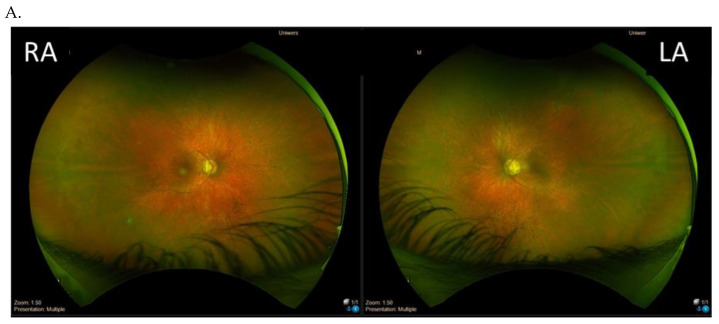
Results of the clinical ophthalmological examinations of both eyes of one patient with a c.2389dup likely pathogenic variant in ORF15 (family 4, patient V, one of a twin). Visual acuity is 0.4 in the right eye (RE) and 0.2 in the left eye (LE). From top to bottom: Optos wide-field color fundus photography (**A**), fundus autofluorescence (**B**), visual field obtained with V4e, III4e and I4e isopters (**C**), and optical coherence tomography of the macula (**D**).

**Table 1 genes-14-01950-t001:** The distinct variants in the 8 index patients with pathogenic and likely pathogenic variants in RPGR gene–exon, classification, class, effect, protein change, novelty, and reference. SNV-single-nucleotide variant. Asterisk means termination.

Patient ID	Family	Nucleotide Change	Exon	Classification	Class	Effect	Protein Change	Novelty	Reference
I.	Family 1	c.2442_2445del	ORF15	deletion	pathogenic	frameshift	p.Gly817fs	no	Branham, 2012 [8] (PMID: 23150612), Vervoort, 2000 [5] (PMID: 10932196), Churchill, 2013 [19] (PMID: 23372056)
II.	Family 2	c.2340_2341del	ORF15	deletion	pathogenic	frameshift	p.Arg780fs	no	Bader, 2003 [7] (PMID: 12657579) Neidhardt, 2008 [11] (PMID: 18552978);
III.	Family 3	c.2587G>T	ORF15	SNV	likely pathogenic	nonsense	p.Glu863*	yes	this study
IV. and V.	Family 4	c.2389dup	ORF15	duplication	likely pathogenic	frameshift	p.Glu797fs	yes	this study
VI.	Family 5	c.2455dup	ORF15	duplication	likely pathogenic	frameshift	p.Val819fs	yes	this study
VII.	Family 6	c.593G>T	6	SNV	likely pathogenic	missense	p.Gly198Val-	yes	this study
VIII.	Family 7	c.799G>C	8	SNV	likely pathogenic	missense	p.Gly267Arg-	yes	this study

**Table 2 genes-14-01950-t002:** Clinical data of the 8 index patients with pathogenic and likely pathogenic variants in the RPGR gene: age at onset, age during the examination, visual acuity at the last examination, kinetic visual field, and optical coherence tomography (OCT) results.

Patient ID	Nucleotide Change	Age at Onset (Years)	Age at the Last Examination (Years)	Visual Acuity at the Examination (Snellen Charts)	Kinetic Visual Field	Optical Coherence Tomography (Central Foveal Thickness in Micrometers)
I (family 1)	c.2442_2445del	10	31	0.7 0.7	Advanced constriction, only V4e remarkable	277 264
II (family 2)	c.2340_2341del	12	42	0.01 0.01	Advanced constriction, only V4e remarkable	208 205
III (family 3)	c.2587G>T	8	36	0.5 0.6	Middle constriction, V4e, III4e and I4e remarkable	143 147
IV (family 4)	c.2389dup	10	22	0.7 0.7	Middle constriction, V4e, III4e remarkable	223 225
V (family 4)	c.2389dup	10	22	0.4 0.2	Middle constriction, V4e, III4e and I4e remarkable	260 240
VI (family 5)	c.2455dup	14	25	0.6 0.3	Middle constriction, V4e, remarkable	205 186
VII (family 6)	c.593G>T	12	20	0.5 0.5	Middle con-striction, V4e, III4e and I4e remarkable	250 245
VIII (family 7)	c.799G>C	6	47	0.2 0.3	Middle con-striction, V4e, III4e and I4e remarkable	240 230

## Data Availability

The data that support the findings of this study are available from the corresponding author upon reasonable request.

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
