# Peer review of "Identification of the RPGR Gene Pathogenic Variants in a Cohort of Polish Male Patients with Retinitis Pigmentosa Phenotype"

_genes, 2023, doi:10.3390/genes14101950_

Round 1
Reviewer 1 Report
The work presented by Nowomiejska et al. summarises the analysis of 45 males affected with RP from Poland. The more interesting results are: the percentage of individuals that RPGR mutations within the cohort, the description of novel mutations in the RPGR gene, and the fact that two dizygotic twins bearing the same hemizygous mutation show a different retinal phenotype in severity and symptoms.
However, the type of variants identified changes in different places of the text and is very confusing.
Overall, The manuscript requires further text editing (for instance, the sections should be in bold font, and the indent in some paragraphs is too large or too small) and the work would benefit from some more explanation/discussion on several points.
GENERAL MAJOR COMMENTS
1) The quality of the images is so low as not allowing the reviewer to really review the results. The quality has to be greatly improved to deserve publication. Neither the pedigrees nor the sequencing electropherograms are readable as they are now. Please, susbtitute them for high quality tiff or png images.
2) The Tables are difficult to interpret because the font size is large and the layering is not adequate for comprehension. If presented in horizontal position, they may improve a lot the understanding of the reader.
SPECIFIC COMMENTS
3) Figures have “figure legends” that are placed at the bottom of each Figure (not on top as they are now). Instead, Tables have the title on top. This is the general convention and the authors should abide to it.
4) Table 1- It is not clear how the author consider a new missense variant as “pathogenic” or a frameshift or nonsense mutation as “likely pathogenic”. They should use the ACMG convention, which is a guide that serves for all the people that works on genetic diagnosis that find a new variant that may be potentially pathogenic. The authors should use the guide and provide accordingly the classification of their variants.
5) In Table 1, family 3 has a NONSENSE mutation, not a missense. Also, case number VIII is numbered again as VII. Please correct.
6) Figure 1, aside the blurriness of the figure, the electropherograms are difficult to interpret, particularly for Variants/Patients I and II, in which the alignment of the consensus sequence with that of the patient is shifted. Moreover, the electropherogram of Patient I shows several peaks in the same point. Please amend and attach precise images with a correct sequence alignment.
7) Line 190, what is a pseudoheterozygous mutation in a male? Maybe the authors meant a mutation in a hemizygotic male? Please, explain.
8) The Results are extremely short, for instance, there is no comment on the phenotype of the patients in the text. Also, the fact that two dizygotic twins show a different retinal phenotype and the type of differences is not explained. Please, lengthen the results so that it relates instead of merely listing tables and figures.
9) Lines 209-210. The verb “compounded” most probably should be “hampered” or “hindered”. The authors mention benign indel polymorphisms in ORF15, but the articles they refer actually do not support this sentence. Please, find the correct references or omit this sequence, which is confusing.
10) Lines 113 to 115, an line 219, the authors have identified 7 different mutations in their cohort. According to Table 1, two frameshift mutations have been previously reported, and of the five new variants, 2 caused a frameshift, one was nonsense, and two were missense. The number and type of variants is different in each sentence as well as in the Table. Please, be consistent all over the text.
The manuscript needs editing, particularly on the quality of the figures. The English needs some minor revision.
Author Response
Dear Reviewer 1,
We are very grateful for the opportunity to submit the revised manuscript and your help in improving it. We strictly followed the Reviewer's suggestions, as is presented below and in the manuscript with track changes.
The work presented by Nowomiejska et al. summarises the analysis of 45 males affected with RP from Poland. The more interesting results are: the percentage of individuals that RPGR mutations within the cohort, the description of novel mutations in the RPGR gene, and the fact that two dizygotic twins bearing the same hemizygous mutation show a different retinal phenotype in severity and symptoms.
Thank you for this comment.
In the abstract it has been added as follows:
Dizygotic twins bearing the same hemizygous mutation showed a different retinal phenotype in severity and symptoms.
However, the type of variants identified changes in different places of the text and is very confusing.
Overall, The manuscript requires further text editing (for instance, the sections should be in bold font, and the indent in some paragraphs is too large or too small) and the work would benefit from some more explanation/discussion on several points.
GENERAL MAJOR COMMENTS
- The quality of the images is so low as not allowing the reviewer to really review the results. The quality has to be greatly improved to deserve publication. Neither the pedigrees nor the sequencing electropherograms are readable as they are now. Please, susbtitute them for high quality tiff or png images.
The pedigrees (figure 1) are presented in another software and have been improved.
The images of the sequencing electropherograms are the SeqScape Software screenshots of the sequencing results analysis. They show the reference sequence the obtained PCR products were compared. The images quality has been improved.
2) The Tables are difficult to interpret because the font size is large and the layering is not adequate for comprehension. If presented in horizontal position, they may improve a lot the understanding of the reader.
Although I tried, I was unable to make the tables horizontal. Could it be done during professional editing of the manuscript prior publication?
SPECIFIC COMMENTS
- Figures have “figure legends” that are placed at the bottom of each Figure (not on top as they are now). Instead, Tables have the title on top. This is the general convention and the authors should abide to it.
Figure legends are now placed at the bottom of the figures.
4) Table 1- It is not clear how the author consider a new missense variant as “pathogenic” or a frameshift or nonsense mutation as “likely pathogenic”. They should use the ACMG convention, which is a guide that serves for all the people that works on genetic diagnosis that find a new variant that may be potentially pathogenic. The authors should use the guide and provide accordingly the classification of their variants.
In the results section it has been written (line ): The ACMG guideline recommends using prediction programs to assess the possible pathogenicity of variants. Five variants identified in our study: nonsense type c.2587 G>T and frameshift c.2455dup according to ACMG were classified as likely pathogenic (criteria: PVS1, PM2), as well as the c.593G>T variant (criteria: PP3, PM2, PM5). Regarding the c.799G>C variant, the word likely was omitted during editing.
In the results section it has been written (line 130-134): In silico analysis (using varSEAK, PolyPhen-2, SIFT) were performed for two patients with a missense mutation. These changes do not affect splicing, but can impact on protein structure and function. Obtained values (PolyPhen-2 – score: 1 ”probably damaging”; SIFT – score: 0,01 deleterious ) corroborate their potential pathogenicity.
The following sentence has been deleted (line 140-142): As for the functional analysis, it was performed in silico using VarSeak (listed in the methodology) for three patients with missense mutations. The analysis showed that the folding of the product is not affected by these changes.
5) In Table 1, family 3 has a NONSENSE mutation, not a missense.
Data in table 1 have been corrected into: nonsense and p.Glu863*.
Also, case number VIII is numbered again as VII. Please correct.-it has been corrected as requested.
6) Figure 1, aside the blurriness of the figure, the electropherograms are difficult to interpret, particularly for Variants/Patients I and II, in which the alignment of the consensus sequence with that of the patient is shifted. Moreover, the electropherogram of Patient I shows several peaks in the same point. Please amend and attach precise images with a correct sequence alignment.
New improved electropherograms have been attached. In electropherograms of patients I and II, the deletion sites are marked with a circle on the reference sequence to facilitate interpretation.
7) Line 190, what is a pseudoheterozygous mutation in a male? Maybe the authors meant a mutation in a hemizygotic male? Please, explain.
It was written in the discussion chapter as it was explained in the cited article (Tracewska, A. M.; Kocyła-Karczmarewicz, B.; Rafalska, A.; Murawska, J.; Jakubaszko-Jabłónska, J.; Rydzanicz, M.; Stawiński, P.; Ciara, E.; Lipska-Ziętkiewicz, B. S.; Khan, M. I.; Cremers, F. P. M.; Płoski, R.; Chrzanowska, K. H. Non-syndromic inherited retinal diseases in Poland: Genes, mutations, and phenotypes Mol Vis. 2021, 16, 27, 457-465.) as follows (line 270-272):
This change was not present in the mother, and most probably represents an early embryonic somatic mutation, present in the peripheral blood and the buccal swab-derived DNA of the patient.
8) The Results are extremely short, for instance, there is no comment on the phenotype of the patients in the text. Also, the fact that two dizygotic twins show a different retinal phenotype and the type of differences is not explained. Please, lengthen the results so that it relates instead of merely listing tables and figures.
In the abstract it is now written (line 23-24):
Dizygotic twins bearing the same hemizygous mutation showed a different retinal phenotype in regard to severity of symptoms.
In the results section it has been added as follows:
Line 186: The main clinical findings of the cohort are included in table 2.
Line 200-204:
Majority of the patients (6 patients apart from twins) presented with typical RP retinal phenotype bone-spicule pigment deposits in the mid peripheral retina, optic disc pallor and RPE atrophy (Figure 3). Also 6 patients demonstrated middle con-striction of the visual field, 2 patients demonstrated advanced constriction of the visu-al field (10 degrees with V4e isopter).
Figures 3,4 and 5 have ben added.
In the discussion chapter it has been added as follows (line 336-338):
The differences in retinal phenotype may be due to incomplete penetrance, thus genotype-phenotype correlations may be very difficult. Moreover, other genes may be involved, thus WGS could be helpful to solve the complete genetic background. Genes affected in RP are multiple and cover different steps of the visual cycle (phototransduction, retinal metabolism, tissue development and maintenance, cellular structure, splicing).
The following reference hs been aded:
- Ferrari, S.; Di Iorio, E.; Barbaro, V., Ponzin, D., Sorrentino, F.S.; Parmeggiani, F. Retinitis pigmentosa: genes and disease mechanisms. Curr Genomics. 2011, 12(4), 238-49.
9) Lines 209-210. The verb “compounded” most probably should be “hampered” or “hindered”. The authors mention benign indel polymorphisms in ORF15, but the articles they refer actually do not support this sentence. Please, find the correct references or omit this sequence, which is confusing.
The following sentence has been deleted (line 291-292): which is compounded further by the presence of many benign polymorphic insertions or deletions (indels).
10) Lines 113 to 115, an line 219, the authors have identified 7 different mutations in their cohort. According to Table 1, two frameshift mutations have been previously reported, and of the five new variants, 2 caused a frameshift, one was nonsense, and two were missense. The number and type of variants is different in each sentence as well as in the Table. Please, be consistent all over the text.
In the results section it is now written (line 115-116):
Among all 45 male patients we identified 7 distinct variants in 8 patients (includ-ing male twins) in the RPGR gene, including 5 variants present in ORF15. Among all of the variants, 5 were novel: 2 duplications and 3 single nucleotide variants (SNV) , 2 deletions have been already described in the literature (table 1). It resulted in a frameshift in 4 variants, 2 missenses and 1 nonsense.
In the discussion chapter it is written (line 302):
We report four frameshift and one nonsense mutation in our series of 5 variants in ORF15, including three new variants within ORF15 region: two frameshift and one nonsense.
Reviewer 2 Report
This is a nicely written report of 45 patients with X-linked retinitis pigmentosa (XLRP) from Poland. The genetics will be interesting as a treatment for XLRP is currently being developed. My only comment is that TABLE 1 does not seem to have aligned columns, which makes it difficult to read. It is probably a type-setting error.
Author Response
Thank you for your positive comment.
Although I tried, I was unable to make corrections to tables. I hope it could be done during professional editing of the manuscript prior publication?
Round 2
Reviewer 1 Report
The authors have correctly addressed my comments and suggestions. There are some minor editing issues that can be corrected with the help of authors and the required assistance of editors, see below..
Editing minor points for Authors
Figure Legends have to be placed at the bottom of Figures, Figure 1 legend is on top, Figure 2 legend is missing. The rest are all right.
Editing queries for Editors
Concerning the Tables 1 and 2 layout, they should be in horizontal to be read properly.
Author Response
Dear Reviewer,
thank you for your valuable comments. See my responses below:
Editing minor points for Authors
Figure Legends have to be placed at the bottom of Figures, Figure 1 legend is on top, Figure 2 legend is missing. The rest are all right.
The position of figure legends has been changed as requested.
Editing queries for Editors
Concerning the Tables 1 and 2 layout, they should be in horizontal to be read properly.
I would like to ask the Editors for correcting the layout of table 1 and 2 in the next steps., as I was not able to perform that.
Your sincerely,
Katarzyna Nowomiejska